# State of the Art in Exocrine Pancreatic Insufficiency

**DOI:** 10.3390/medicina56100523

**Published:** 2020-10-07

**Authors:** Carmelo Diéguez-Castillo, Cristina Jiménez-Luna, Jose Prados, José Luis Martín-Ruiz, Octavio Caba

**Affiliations:** 1Department of Gastroenterology, San Cecilio University Hospital, 18012 Granada, Spain; carmelo89dc@gmail.com (C.D.-C.); jlmartin@ugr.es (J.L.M.-R.); 2Institute of Biopathology and Regenerative Medicine (IBIMER), University of Granada, 18100 Granada, Spain; crisjilu@ugr.es (C.J.-L.); jcprados@ugr.es (J.P.)

**Keywords:** exocrine pancreatic insufficiency, prevalence, clinic relevance, diagnosis, treatment, pancreatic enzyme replacement therapy

## Abstract

Exocrine pancreatic insufficiency (EPI) is defined as the maldigestion of foods due to inadequate pancreatic secretion, which can be caused by alterations in its stimulation, production, transport, or interaction with nutrients at duodenal level. The most frequent causes are chronic pancreatitis in adults and cystic fibrosis in children. The prevalence of EPI is high, varying according to its etiology, but it is considered to be underdiagnosed and undertreated. Its importance lies in the quality of life impairment that results from the malabsorption and malnutrition and in the increased morbidity and mortality, being associated with osteoporosis and cardiovascular events. The diagnosis is based on a set of symptoms, indicators of malnutrition, and an indirect non-invasive test in at-risk patients. The treatment of choice combines non-restrictive dietary measures with pancreatic enzyme replacement therapy to correct the associated symptoms and improve the nutritional status of patients. Non-responders require the adjustment of pancreatic enzyme therapy, the association of proton pump inhibitors, and/or the evaluation of alternative diagnoses such as bacterial overgrowth. This review offers an in-depth overview of EPI in order to support the proper management of this entity based on updated and integrated knowledge of its etiopathogenesis, prevalence, diagnosis, and treatment.

## 1. Introduction

Exocrine pancreatic insufficiency (EPI) is defined in the most recent reviews as the inability of the pancreas to secrete enzymes and bicarbonate for action on the intestinal lumen to accomplish the normal digestion of food [1,2,3,4,5,6,7,8]. EPI can be caused by an alteration at any point in the digestive chain in which the exocrine pancreas is involved [1,4,6,7]. Alterations have been described in the following processes:(a)Pancreatic stimulation: Insufficient activation of pancreatic secretion can be caused by diseases (e.g., celiac disease) that reduce the release of cholecystokinin (CCK) from the duodenal mucosa or by pancreatic/gastrointestinal surgery. EPI can also result from the treatment of neuroendocrine tumors with somatostatin analogs, given that somatostatin is a physiological pancreatic secretion inhibitor [3,9].(b)Pancreatic juice synthesis: Damage to the pancreatic parenchyma reduces the production and secretion of pancreatic enzymes by acinar cells and of bicarbonate by pancreatic ducts. This functional loss can be caused by various diseases, including chronic pancreatitis, cystic fibrosis, pancreatic cancer, or acute necrotizing pancreatitis, or by pancreatic resection for the treatment of some of these.(c)Pancreatic juice transport: Obstruction of the passage of pancreatic juice through the pancreatic duct prevents its arrival into the intestinal lumen to carry out its digestive activity. This problem can be caused by disorders such as cystic fibrosis, with the production of a thicker secretion, or by various types of pancreatic tumor.(d)Synchronization of gastrointestinal secretions: Asynchrony in the interaction of nutrients with biliopancreatic secretions results in the incorrect digestion of foods. This problem is usually caused by anatomical changes produced by pancreatobiliary or gastrointestinal surgery. This phenomenon is also observed in patients with Crohn’s disease or short bowel syndrome. Less frequently, there is no enzyme activation despite adequate secretion and the arrival of pancreatic juice to the duodenum, as in cases of hyperchlorhydria.

According to the above data, the terms exocrine pancreatic secretion and function should not be considered synonymous, given the possibility of EPI when the secretion is adequate, as in disorders that alter the interaction between secretion and food [1,3].

EPI is almost always produced by a pancreatic disease, with the most frequent cause being chronic pancreatitis in adults [10,11] and cystic fibrosis in children [4]. Other etiologies include acute pancreatitis [12,13], pancreatic tumors [14], diabetes mellitus [15,16], celiac disease, inflammatory bowel disease [17,18], gastrointestinal and pancreatic surgery [19,20,21], HIV and genetic and congenital factors. Some of these causes have are less frequent, and there is even debate around their true role in the etiology of EPI.

Capurso et al. (2019) divided the causes of EPI between pancreatic and extra-pancreatic disorders (Table 1) [3], whereas other authors distinguished between those that had clearly demonstrated a connection with EPI and those that had not, assisting the establishment of a definitive diagnosis and the definition of risk groups (Table 2) [5]. In this way, the Australian Pancreas Group stratified the causes of EPI as definite, possible or unlikely, with the latter including type 2 diabetes, celiac disease, inflammatory bowel disease, irritable bowel syndrome, advanced age with weight loss, and intestinal resection [6].

The aim of this review was to summarize and integrate recent knowledge on the etiopathogenesis, prevalence, treatment, and diagnosis of EPI and propose a novel algorithm for the clinical management of this entity, stratifying the risk of developing EPI according to the etiology and exploring the diagnostic usefulness of combinations of variables/symptoms. 

Five electronic databases were searched: Ovid MEDLINE (1946 to present), Embase (1980 to present), Cochrane Library (1979 to present), AMED (1985 to present), and CINAHL (1981 to present). Two groups of search terms were combined, one related to EPI (“exocrine pancreatic insufficiency”, “pancreatic insufficiency”) and the other to the different issues under study (“concept”; “pathogeny”; “etiology”; “prevalence”; “clinical manifestations”; “diagnostic”; “treatment”). The search results were screened for eligibility in an independent and standardized manner by two investigators (C.D.-C., and C.J.-L.), resolving discrepancies by mutual consensus. Titles and abstracts of eligible studies were first screened, followed by the full article. No date, study design, or language limitations were applied in any search. The latest search was conducted on 1 August 2020. However, given that almost all selected studies focused on a specific aspect of EPI and showed a high variability, it was decided that the quantitative synthesis of data in a meta-analysis would not be appropriate.

## 2. Prevalence and Clinical Relevance

The current prevalence of EPI is unknown [3] and is highly variable, due to its multiple etiologies (Table 3) [4]. However, there is general consensus that it is a frequent disease that is underdiagnosed and undertreated [1].

The maldigestion produced by EPI can result in steatorrhea, abdominal distension, flatulence, and/or weight loss. These symptoms of malabsorption are non-specific, and there are even patients who reduce their intake of the foods they tolerate least well and may be paucisymptomatic. The persistence of symptoms impairs the quality of life of patients [1,4,6].

Alterations in intestinal ecology appear to be implicated in pancreatic maldigestion, according to observations of an increase in intestinal inflammation markers in patients with EPI, augmenting intestinal permeability and favoring bacterial overgrowth, thereby contributing to the persistence of symptoms [24].

The importance of EPI also lies in the resulting predisposition to develop malnutrition, which does not differ from that produced by other causes [7], being associated with deficiencies in proteins, oligoelements (Mg, Zn, etc.) and liposoluble vitamins (A, D, E, K) [1]. Malnutrition can lead to the onset of sarcopenia, which is not necessarily accompanied by weight loss. Indeed, Shintakuya et al. (2017) reported that the majority of EPI patients with sarcopenia in their study were overweight [25].

A frequent comorbidity in patients with EPI is osteopenia or osteoporosis, associated with non-traumatic fractures. For this reason, a bone mineral density test is recommended at the diagnosis and periodically thereafter [5]. Malnutrition also reduces the immunocompetence of patients and increases their risk of cardiovascular events. In patients with pancreatic cancer, the development of EPI is a factor for a poor prognosis, and its treatment independently improves survival outcomes [26,27]. In addition, EPI is a cardiovascular risk factor independent of other known factors (arterial hypertension, diabetes, tobacco, obesity), being influenced by the associated malnutrition [28]. The presence of these comorbidities increases the morbidity and mortality of patients with EPI [29].

## 3. Diagnosis

The diagnosis of EPI is currently based on a set of symptoms of maldigestion/malabsorption, on indicators of malnutrition, and on the result of a non-invasive pancreatic test [1]. The combination of two of these criteria should be considered sufficient for the diagnosis and for initiating treatment of pancreatic maldigestion. Nevertheless, when a diagnosis of EPI appears unlikely, it is recommended that a positive indirect test of pancreatic function should be one of the diagnostic criteria applied (Figure 1).

It involves the identification of at-risk patients with pancreatic or extra-pancreatic disorders that are potentially responsible for EPI [5]. There is a high likelihood that a diagnosis of EPI can be established before a diagnostic test is performed in patients with defined causes of EPI, e.g., chronic pancreatitis, acute pancreatitis with extensive necrosis, cancer in the head of the pancreas, or gastrointestinal surgery [1].

It is important to take account of the demonstration over recent years that the presence of postprandial fatty stools is highly predictive of steatorrhea and that their presence combined with weight loss greater than 10% is strongly suggestive of EPI [30,31]. Although symptoms of maldigestion–malabsorption support the diagnosis of EPI, they may be absent or appear later [7].

Evaluation of the nutritional status of patients is based on anthropometric and analytical parameters, including biochemical results for oligoelements, liposoluble vitamins, and lipoproteins. Some authors have proposed a decrease in magnesium, albumin, prealbumin and retinol-binding protein levels as a predictive model for the diagnosis of EPI and for monitoring the response to treatment [32].

Various direct and indirect methods are available to quantify exocrine pancreatic secretions (Table 4). Direct techniques include the quantification of duodenal juice collected by probe or endoscopy after stimulation with secretagogues (secretin or CCK), whereas one indirect approach is to measure the results of this stimulation by detecting metabolites derived from the action of pancreatic enzymes [3,7].

Direct methods offer high sensitivity and specificity but are invasive, expensive, and time-consuming. They are not readily available, and variability among centers has prevented their standardization. For these reasons, they are not useful for post-treatment follow-up. Conversely, indirect methods lack sensitivity in initial stages but are less costly and easier to apply [3,4]. The absence of a widely available, precise, and readily-reproducible diagnostic technique has contributed to the underdiagnosis of EPI [5].

The stool elastase test is the most widely applied non-invasive approach in routine clinical practice. Among multiple advantages, it does not need the collection of feces or a specific pre-test diet, it can be applied at all care levels, and it does not require the suspension of pancreatic enzyme replacement therapy (PERT) because it shows no cross-reaction with the porcine enzymes used in this treatment. Furthermore, it has demonstrated high sensitivity for advanced stages (elastase < 100 μg/g feces = severe EPI). Nevertheless, this method is not without limitations, which must be understood to achieve an optimal result, because many can be easily corrected or modified to minimize false positives [35]. The following points should be taken into consideration:(a)There is no consensus on the cutoff point for an EPI diagnosis, which has generally been arbitrarily considered as <200 μg/g. Nevertheless, the smaller the concentration, the greater the likelihood of EPI, and very high sensitivity and specificity values have been reported for concentrations <15 μg/g. Conversely, EPI is ruled out by very high elastase values (>500 μg/g) [7].(b)The interpretation of borderline elastase values (100–200) should be guided by the observation of symptoms and malnutrition indicators [1]. A closer follow up should be considered, especially if there is persistence of the etiological factors most frequently associated with pancreatic damage (tobacco and alcohol).(c)Samples with a liquid consistency (Bristol scale of 5–7) should be excluded.(d)Potential causes of false positives should be ruled out, including bacterial overgrowth [8]. They may also be attributable to advanced age or conditions such as chronic kidney disease, although these associations have not been fully elucidated.

## 4. Treatment

Despite the evident benefits of initiating treatment, diagnostic tests are carried out in a very small percentage of cases and few patients are treated, often with inadequate doses [36,37,38,39]. Patients with a diagnosis of EPI are given dietary recommendations and pancreatic enzyme replacement therapy (PERT). Their diet should be balanced, with no need to follow fat-restricted or fiber-rich diets. Patient are advised to consume a larger number of smaller-sized meals to facilitate digestion. Liposoluble vitamin supplements can be administered when considered necessary. 

The dose of PERT should be individualized in accordance with the severity of patients and their needs [1]. The minimum dose recommended for PERT is 50,000 Ph.Eur.U. for main meals and half of this dose for snacks, although higher doses should be prescribed in patients with pancreatic cancer or after pancreatic/gastrointestinal surgery. Half of the dose is taken at the beginning of each meal (after a few mouthfuls of food) and the rest at the end. Adjustment of the dose and regimen should take account of poor therapeutic adherence as well as inactivation by gastric acids, improving the response by associating proton-pump inhibitors. If no response is observed, other possible diagnoses should be considered, such as bacterial overgrowth, celiac disease, or a neoplasm (Figure 2) [2].

Bacterial overgrowth has frequently been observed in patients with chronic pancreatitis [40]. Its involvement in the persistence of steatorrhea in adequately supplemented patients is attributed to the malabsorption of bile acids and the alteration of intestinal permeability. However, PERT itself may influence the composition of intestinal flora, inducing the colonization of beneficial bacteria [41] and the association of proton pump inhibitors corrects the low pH, improving the absorption of bile acids.

Hence, the treatment of these patients is currently based on the correction of pancreatic insufficiency using pancreatic extracts and the improvement of duodenal pH to achieve their optimal effectiveness. However, other factors possibly implicated in maldigestion and malabsorption, such as changes in intestinal ecology, can predispose patients to intestinal inflammation and poor bile acid absorption. These should be considered as potential therapeutic targets, administering supplementation with bile acids, prebiotics, probiotics, or other drugs to protect and strengthen the intestinal barrier [24]. In this regard, Hamada et al. (2018) addressed the relationship of certain strains of intestinal microbiota with a worsening of malabsorption symptoms [42]. In addition, experimental rat studies have drawn attention to the intestinal microbiota as a possible therapeutic target [43,44].

In our view, the optimization of PERT, through an improvement in adherence, dose adjustments, and association with proton pump inhibitors, is the cornerstone of treatment, obviating the previously recommended need for dietary fat restrictions or for vitamin or oligoelement supplementation, unless there is a severe deficit that generates symptoms.

Knowledge of new mechanisms underlying maldigestion/malabsorption, such as those that may involve the intestinal microbiota, can help to identify novel therapeutic targets and may represent a change in the approach to EPI [45,46]. Thus, these aspects could change from being the third step in monitoring the effectiveness of therapy in non-responders to being part of first-line treatment, so that the different factors implicated in EPI are treated in a combined manner from the outset.

The effectiveness of PET is indicated by the normalization of malabsorption symptoms and the recovery of nutritional status according to anthropometric and biochemical parameters [7,8], which should be followed up at least once a year [4]. Symptom control improves the quality of life of the patients and nutritional status recovery reduces their mortality risk. PET can ameliorate malabsorption symptoms such as meteorism but has not proved able to relieve abdominal pain.

In patients with pancreatic cancer, PERT has been found to independently improve their survival outcomes to a similar degree to that observed for surgery or chemotherapy [47]. PERT is a cornerstone of the treatment of patients with chronic pancreatitis, alongside a balanced diet, and achieves the correct management of >80 % of patients, with only 10–15 % needing oral nutritional supplements [7].

## 5. Conclusions

Exocrine pancreatic insufficiency is a highly prevalent entity that is underdiagnosed and undertreated. It is essential to achieve an early diagnosis and treatment, not only in patients with well-known causes of this condition (e.g., chronic pancreatitis, cystic fibrosis, pancreatic cancer, or acute pancreatitis with extensive necrosis) but also in those with emerging causes. Diagnosis is usually based on a set of symptoms, undernourishment data, and a noninvasive diagnostic test in at-risk groups. The appropriate implementation of PERT improves the quality of life of these patients by controlling their symptoms and reduces their morbidity and mortality risk by rectifying their nutritional status.

## Figures and Tables

**Figure 1 medicina-56-00523-f001:**
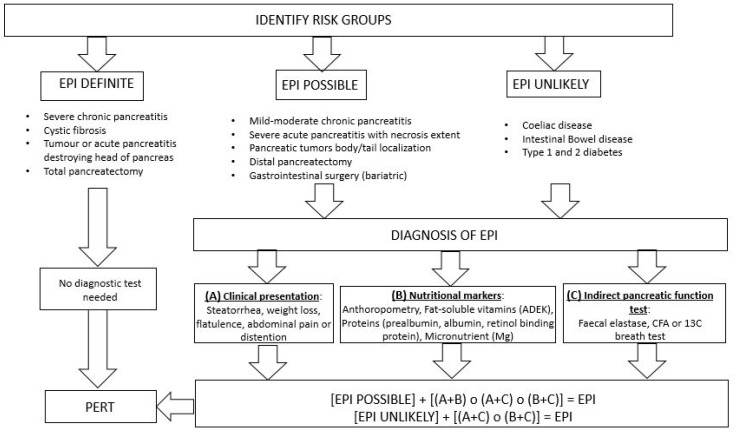
Summary diagnostic algorithm (adapted from Dominguez-Muñoz and the Australian Pancreatic Club). EPI: exocrine pancreatic insufficiency CFA: coefficient of fat absorption 13C: 13C-mixed triglyceride PERT: pancreatic enzymatic replacement therapy.

**Figure 2 medicina-56-00523-f002:**
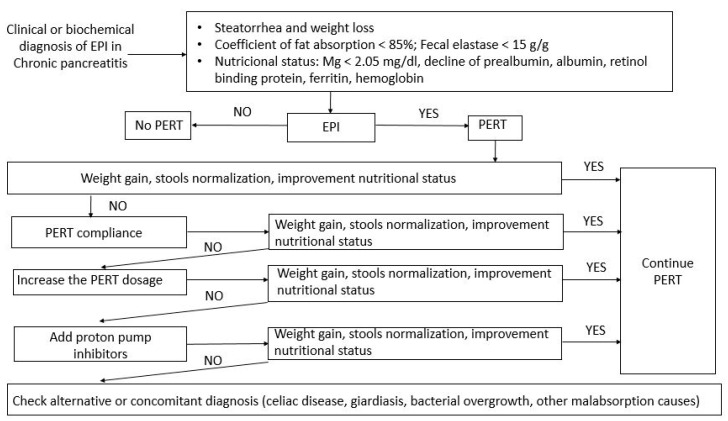
Algorithm for the follow-up of pancreatic enzymatic replacement therapy (PERT) in exocrine pancreatic insufficiency (EPI) (modified from Pezzilli et al., 2013) [34].

**Table 1 medicina-56-00523-t001:** Pancreatic and extra-pancreatic causes of exocrine pancreatic insufficiency (EPI) (adapted from Capurso et al., 2019).

Pancreatic Disorders	Extra-Pancreatic Disorders
Chronic Pancreatitis	Types 1 and 2 diabetes
Cystic Fibrosis	Gastrointestinal Surgery
Pancreatic Tumors	Celiac Disease
Acute Pancreatitis	Inflammatory Bowel Disease
Pancreatic Surgery	HIV
Shwachman–Diamond Syndrome	Sjögren’s Syndrome
Johanson–Blizzard Syndrome	Intestinal Transplant at Pediatric Age
Pancreatic Hemochromatosis [22]	Treatment with Somatostatin Analog
Trypsinogen or Enteropeptidase Deficiency	Advanced Age
	Tobacco Habit [23]

**Table 2 medicina-56-00523-t002:** Defined and possible causes of EPI (adapted from Forsmark).

Defined Causes	Likely Causes
Chronic Pancreatitis	Gastrointestinal Surgery
Cystic Fibrosis	Tobacco Habit [23]
Pancreatic Surgery	Types 1 and 2 Siabetes
Pancreatic Tumor/Cancer	Celiac Disease
Benign Main Pancreatic Duct Obstruction	Zollinger-Ellison Syndrome (gastrinoma)
Shwachman-Diamond Syndrome	HIV
Johanson-Blizzard Syndrome	Advanced Age
Hemochromatosis [22]	Severe Malnutrition
	Acute Pancreatitis without Severe or Recurrent Necrosis

**Table 3 medicina-56-00523-t003:** Prevalence of EPI according to its etiology.

Cause of EPI	Prevalence
Chronic Pancreatitis	At the Diagnosis: 10% [7] After 10–12 Years with the Disease: 60–90% [4]
Acute Pancreatitis [2]	Short-Term: 60% Long-Term: 33%
Pancreatic Tumor [2]	Unresectable: 90% Resectable: - Pre-Surgery: 20–44% - Post-Surgery: 60%
Pancreatic Surgery [3]	Whipple Procedure: 85–95% Distal Pancreatectomy: 20–50%
Cystic Fibrosis [2]	85% (the Majority at Birth)
Gastrointestinal Surgery [3]	Total/Subtotal Gastrectomy: 40–80% Esophagectomy: 16%
Type 1 Diabetes	Severe: 10–30%; Mild–Moderate: 22–56% [4] 40% [2]
Type 2 Diabetes	5–46 % [4] 27 % [2]
Celiac Disease	5–80% [3]
Inflammatory Bowel Disease	14–74 % [4]
HIV	26–45% [4]

**Table 4 medicina-56-00523-t004:** Diagnostic tests available to quantify exocrine pancreatic secretion (modified from Afghani et al., 2014; Pezzilli et al., 2013) [33,34]

Diagnostic test type	Advantages	Disadvantages
Direct		
Duodenal Intubation Test	High Sensitivity and Specificity Preferably with Secretin Stimulation and Measurement of Bicarbonate	Invasive Laborious Expensive Only Available in Specialist Centers
Endoscopic Test		
Indirect		
13C-mixed triglyceride Breath Test	Simple Noninvasive Useful to Monitor the Response to PERT	Prolonged Time (6 hours) Not Widely Available False Positives (Malabsorption of Non-Pancreatic Fats and Liver or Lung Disease) Non-specific Low Sensitivity for Mild EPI
Stool Elastase	Very Simple Noninvasive Fast Widely Available	Not Useful for PERT Monitoring Low Sensitivity for Mild Cases False Positives (Liquid Stools, Intestinal Inflammation) Not Reliable After Pancreatic Resection
Fat absorption Coefficient	Gold Standard	Difficult Application/Adherence Only for Severe EPI Cases Diet of 100g Fats for 5 Days Gathering Stools for 3 Days False Positives (Malabsorption of Non-Pancreatic Fats)

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
