# Peer review of "State of the Art in Exocrine Pancreatic Insufficiency"

_medicina, 2020, doi:10.3390/medicina56100523_

Round 1
Reviewer 1 Report
Thank you for your submission which is a fairly comprehensive overview of pancreatic exocrine insufficiency.
This submission discusses the background/rationale underlying pancreatic exocrine insufficiency and then systematically discusses its importance, etiologies, diagnosis and treatments.
Reviews of this kind in this field are very important as it is well known that this entity is underrecognized and undertreated with significant consequences for patients as the authors cite in several of their references.
I believe this submission is easy to read and provides a good synopsis on the topic. I plan to show it to my students and residents in teaching rounds and to my collegues in surgery and internal medicine for example if it is published.
I found this to be a well written overview of pancreatic exocrine insufficiency and did not find any major content areas. There are a few mild stylistic English language issues but as a native English speaker they did not at all detract from the meaning of the article and I found it easy to read overall.
LIne 86 needs to be reworded. The term "worst" needs to be changed or the syntax/diction of the sentence needs to be changed."
Author Response
Response 1: We are grateful for these positive remarks. Line 86 has been reworded accordingly (now line 98).
Reviewer 2 Report
This is a narrative review on the diagnosis and tretment of exocrine pancreatic insufficiency similat to many other present in literature.
Probably the paper may be improved and may be more interesting as systematic review.
In this case the authors should add a a section reporting the search strategy.
Especially regarding the diagnosis of pancreatic maldigestion the references should be updated: for example there are new possibility to fiagnose clinically the pancreatic maldigestion and this should be added to the text (Pezzilli R. Applicability of a checklist for the diagnosis and treatment of severe exocrine pancreatic insufficiency: a survey on the management of pancreatic maldigestion in Italy. Panminerva Med. 2016;58(4):245-252. Pezzilli R, Capurso G, Falconi M, et al. The Applicability of a Checklist for the Diagnosis and Treatment of Exocrine Pancreatic Insufficiency: Results of the Italian Exocrine Pancreatic Insufficiency Registry. Pancreas. 2020;49(6):793-798.).
The authors should also clear indicate when we treat the pancreatic maldigestion: In other words the treatment should be driven by the results of laboratory tests, on the results of clinical signs or a a combination: this is the cornestone of highly debated issue.
In addition, a section of the the effects of pancreatic maldigestion of intestinal ecology shoul ad be considered. In fact, the relationships between pancreatic maldigestion, intestinal ecology and intestinal inflammation have not received particular attention, even if in clinical practice these mechanisms may be responsible for the low efficacy of pancreatic extracts in abolishing steatorrhea in some patients. The best treatments for pancreatic maldigestion should be re-evaluated, taking into account not only the correction of pancreatic insufficiency using pancreatic extracts and the best duodenal pH to permit optimal efficacy of these extracts, but we also need to consider other therapeutic approaches including the decontamination of intestinal lumen, supplementation of bile acids and, probably, the use of probiotics which may attenuate intestinal inflammation in chronic pancreatitis patients (Pezzilli R. Chronic pancreatitis: maldigestion, intestinal ecology and intestinal inflammation. World J Gastroenterol. 2009;15(14):1673-1676. doi:10.3748/wjg.15.1673)
The point of view of the authors on the various treatment options is lacking.
The Table 4 should be revised because chymotrypsinogen an serum trypsinogen are no longer used in clinical practice as well as direct fuction tests. Using these latter tests we should treat also patients with severe exocrine pancreatic insufficiency or only those with clinical and biochemical signs: this is the dilemma.
Figure 2 is quite consusing and dhoul be betted represented apecially regarding the possibility that a combination of clinical and biochemical signs drive to the treatment.
Author Response
Point 1: This is a narrative review on the diagnosis and treatment of exocrine pancreatic insufficiency similar to many other present in literature.
Response 1: Previous reviews on exocrine pancreatic insufficiency have focused on specific aspects, such as the etiology or diagnosis. We believe that our review offers a novel contribution to the literature because it provides a comprehensive and in-depth overview of this entity and addresses all associated aspects, which are strongly inter-related.
Point 2: Probably the paper may be improved and may be more interesting as systematic review. In this case the authors should add a a section reporting the search strategy.
Response 2: This information has been added in the revised Introduction, as follows:
“Five electronic databases were searched: Ovid MEDLINE (1946 to present), Embase (1980 to present), Cochrane Library (1979 to present), AMED (1985 to present), and CINAHL (1981 to present). Two groups of search terms were combined, one related to EPI (“exocrine pancreatic insufficiency”, “pancreatic insufficiency”) and the other to the different issues under study (“concept”; “pathogeny”; “etiology”; “prevalence”; “clinical manifestations”; “diagnostic”; “treatment”). The search results were screened for eligibility in an independent and standardized manner by two investigators (C.D-C., and C.J-L.), resolving discrepancies by mutual consensus. Titles and abstracts of eligible studies were first screened, followed by the full article. No date, study design, or language limitations were applied in any search. The latest search was conducted on 01/08/2020. In addition to the search results, references in eligible studies were screened and included when review eligibility criteria were met. However, given that almost all selected studies focused on a specific aspect of EPI and showed a high variability, it was decided that the quantitative synthesis of data in a meta-analysis would not be appropriate.” (Page 2; line 76 to 87).
Point 3: Especially regarding the diagnosis of pancreatic maldigestion the references should be updated: for example there are new possibility to fiagnose clinically the pancreatic maldigestion and this should be added to the text.
Response 3: We have updated the references and added a comment on the new possibility to clinically diagnose pancreatic maldigestion, as recommended:
“It is important to take account of the demonstration over recent years that the presence of postprandial fatty stools is highly predictive of steatorrhea and that their presence combined with weight loss greater than 10% is strongly suggestive of EPI [30,31].” (Page 5; line 134 to 136).
In addition, given the importance of these issues, we have modified Figure 1 on the diagnosis of EPI so that both steatorrhea and weight low are included as clinical criteria.
- Pezzilli, R. Applicability of a checklist for the diagnosis and treatment of severe exocrine pancreatic insufficiency: a survey on the management of pancreatic maldigestion in Italy. Panminerva Med 2016, 58, 245-252.
- Pezzilli, R.; Capurso, G.; Falconi, M.; Frulloni, L.; Macarri, G.; Costamagna, G.; Di Leo, A.; Salacone, P.; Carroccio, A.; Zerbi, A. The Applicability of a Checklist for the Diagnosis and Treatment of Exocrine Pancreatic Insufficiency: Results of the Italian Exocrine Pancreatic Insufficiency Registry. Pancreas 2020, 49, 793-798.
Point 4: The authors should also clear indicate when we treat the pancreatic maldigestion: In other words the treatment should be driven by the results of laboratory tests, on the results of clinical signs or a a combination: this is the cornestone of highly debated issue.
Response 4: As requested by the reviewer, we have added the following new paragraph in the revised Diagnosis section:
“The diagnosis of EPI is currently based on a set of symptoms of maldigestion/malabsorption, on indicators of malnutrition, and on the result of a non-invasive pancreatic test [1]. The combination of two of these criteria should be considered sufficient for the diagnosis and for initiating treatment of pancreatic maldigestion. Nevertheless, when a diagnosis of EPI appears unlikely, it is recommended that a positive indirect test of pancreatic function should be one of the diagnostic criteria applied (Figure 1).” (Page 4; line 119 to 124).
Point 5: In addition, a section of the effects of pancreatic maldigestion of intestinal ecology should be considered. In fact, the relationships between pancreatic maldigestion, intestinal ecology and intestinal inflammation have not received particular attention, even if in clinical practice these mechanisms may be responsible for the low efficacy of pancreatic extracts in abolishing steatorrhea in some patients.
Response 5: We fully agree with the reviewer and have now added the following text on this important issue in the Diagnosis and Treatment sections, respectively:
“Alterations in intestinal ecology appear to be implicated in pancreatic maldigestion, according to observations of an increase in intestinal inflammation markers in patients with EPI, augmenting intestinal permeability and favoring bacterial overgrowth, thereby contributing to the persistence of symptoms [24]”. (Page 3 and 4; line 100 to 103).
“Bacterial overgrowth has frequently been observed in patients with chronic pancreatitis [40]. Its involvement in the persistence of steatorrhea in adequately supplemented patients is attributed to the malabsorption of bile acids and the alteration of intestinal permeability. However, PERT itself may influence the composition of intestinal flora, inducing the colonization of beneficial bacteria [41] and the association of proton pump inhibitors corrects the low pH, improving the absorption of bile acids.” (Page 7; lines 197 to 202).
- Pezzilli, R. Chronic pancreatitis: maldigestion, intestinal ecology and intestinal inflammation. World J Gastroenterol 2009, 15, 1673-1676.
- Akshintala, V.S.; Talukdar, R.; Singh, V.K.; Goggins, M. The Gut microbiome in pancreatic disease. Clin Gastroenterol Hepatol 2019, 17, 290-295.
- Nishiyama, H.; Nagai, T.; Kudo, M.; Okazaki, Y.; Azuma, Y.; Watanabe, T.; Goto, S.; Ogata, H.; Sakurai, T. Supplementation of pancreatic digestive enzymes alters the composition of intestinal microbiota in mice. Biochem Biophys Res Commun 2018, 495, 273-279.
Point 6: The best treatments for pancreatic maldigestion should be re-evaluated, taking into account not only the correction of pancreatic insufficiency using pancreatic extracts and the best duodenal pH to permit optimal efficacy of these extracts, but we also need to consider other therapeutic approaches including the decontamination of intestinal lumen, supplementation of bile acids and, probably, the use of probiotics which may attenuate intestinal inflammation in chronic pancreatitis patients (Pezzilli R. Chronic pancreatitis: maldigestion, intestinal ecology and intestinal inflammation. World J Gastroenterol. 2009;15(14):1673-1676. doi:10.3748/wjg.15.1673).
Response 6: We are grateful for this suggestion and have added the following in the revised Treatment section:
“Hence, the treatment of these patients is currently based on the correction of pancreatic insufficiency using pancreatic extracts and the improvement of duodenal pH to achieve their optimal effectiveness. However, other factors possibly implicated in maldigestion and malabsorption, such as changes in intestinal ecology, can predispose patients to intestinal inflammation and poor bile acid absorption. These should be considered as potential therapeutic targets, administering supplementation with bile acids, prebiotics, probiotics, or other drugs to protect and strengthen the intestinal barrier [24]. In this regard, Hamada et al. (2018) addressed the relationship of certain strains of intestinal microbiota with a worsening of malabsorption symptoms [42]. In addition, experimental rat studies have drawn attention to the intestinal microbiota as a possible therapeutic target [43,44].” (Page 7; line 203 to 222).
- Pezzilli, R. Chronic pancreatitis: maldigestion, intestinal ecology and intestinal inflammation. World J Gastroenterol 2009, 15, 1673-1676.
- Hamada, S.; Masamune, A.; Nareshima, T.; Simosegawa, T. Differences in Gut Microbiota profiles between autoimmune pancreatitis and chronic pancreatitis. Tohoku J Exp Med 2018, 244, 113-117.
- Hu, Y.; Teng, C.; Yu, S.; Wang, X.; Liang, J.; Bai, X.; Dong, L.; Song, T.; Yu, M.; Qu, J. Inonotus obliquus polysaccharide regulates gut microbiota of chronic pancreatitis in mice. AMB Expr 2017, 7, 39.
- Li, K.; Zhuo, C.; Teng, C.; Yu, S.; Wang, X.; Hu, Y.; Ren, G.; Yu, M.; Qu, J. Effects of Ganoderma lucidum polysaccharides on chronic pancreatitis and intestinal microbiota in mice. Int J Biol Macromol 2016, 93, 904-912.
Point 7: The point of view of the authors on the various treatment options is lacking.
Response 7: This now provided in the revised manuscript, as follows:
“In our view, the optimization of PERT, through an improvement in adherence, dose adjustments, and association with proton pump inhibitors, is the cornerstone of treatment, obviating the previously recommended need for dietary fat restrictions or for vitamin or oligoelement supplementation, unless there is a severe deficit that generates symptoms.
Knowledge of new mechanisms underlying maldigestion/malabsorption, such as those that may involve the intestinal microbiota, can help to identify novel therapeutic targets and may represent a change in the approach to EPI [45,46]. Thus, these aspects could change from being the third step in monitoring the effectiveness of therapy in non-responders to being part of first-line treatment, so that the different factors implicated in EPI are treated in a combined manner from the outset.” (Page 7; line 213 to 222).
- Leal-Lopes, C.; Velloso, F.J.; Campopiano, J.C.; Sogayar, M.C.; Correa, R.G. Roles of comensal microbiota in pancreas homeostasis and pancreatic pathologies. J Diabetes Res 2015, 2015, 284680.
- Memba, R.; Duggan, S.N.; Ni Chonchubhair, H.M.; Griffin, O.M.; Bashir, Y.; O'Connor. D.B.; Murphy, A.; McMahon, J.; Volcov, Y.; Ryan, B.M.; Conlon, K.C. The potential role of gut microbiota in pancreatic disease: A systematic review. Pancreatology 2017, 17, 867-874.
Point 8: The Table 4 should be revised because chymotrypsinogen and serum trypsinogen are no longer used in clinical practice as well as direct fuction tests. Using these latter tests we should treat also patients with severe exocrine pancreatic insufficiency or only those with clinical and biochemical signs: this is the dilemma.
Response 8: We agree with the reviewer and have removed chymotrypsinogen and serum trypsinogen tests from table 4.
Point 9: Figure 2 is quite confusing and dhoul be betted represented especially regarding the possibility that a combination of clinical and biochemical signs drive to the treatment.
Response 9: The Figure has been modified accordingly.
Round 2
Reviewer 2 Report
I have no further comments